# Antibacterial Screening of Isoespintanol, an Aromatic Monoterpene Isolated from *Oxandra xylopioides* Diels

**DOI:** 10.3390/molecules27228004

**Published:** 2022-11-18

**Authors:** Orfa Inés Contreras Martínez, Alberto Angulo Ortíz, Gilmar Santafé Patiño

**Affiliations:** 1Biology Department, Faculty of Basic Sciences, University of Córdoba, Montería 230002, Colombia; 2Chemistry Department, Faculty of Basic Sciences, University of Córdoba, Montería 230002, Colombia

**Keywords:** nosocomial infection, isoespintanol, *Oxandra xylopioides*, antibacterial activity, antibiofilms, *Pseudomonas aeruginosa*

## Abstract

The incidence of nosocomial infections, as well as the high mortality and drug resistance expressed by nosocomial pathogens, especially in immunocompromised patients, poses significant medical challenges. Currently, the efficacy of plant compounds with antimicrobial potential has been reported as a promising alternative therapy to traditional methods. Isoespintanol (ISO) is a monoterpene with high biological activity. Using the broth microdilution method, the antibacterial activity of ISO was examined in 90 clinical isolates, which included 14 different species: (*Escherichia coli* (38), *Pseudomonas aeruginosa* (12), *Klebsiella pneumoniae* (13), *Acinetobacter baumannii* (3), *Proteus mirabilis* (7), *Staphylococcus epidermidis* (3), *Staphylococcus aureus* (5), *Enterococcus faecium* (1), *Enterococcus faecalis* (1), *Stenotrophomonas maltophilia* (2), *Citrobacter koseri* (2), *Serratia marcescens* (1), *Aeromonas hydrophila* (1), and *Providencia rettgeri* (1). MIC_90_ minimum inhibitory concentration values ranged from 694.3 to 916.5 µg/mL and MIC_50_ values from 154.2 to 457.3 µg/mL. The eradication of mature biofilms in *P. aeruginosa* after 1 h of exposure to ISO was between 6.6 and 77.4%, being higher in all cases than the percentage of biofilm eradication in cells treated with ciprofloxacin, which was between 4.3 and 67.5%. ISO has antibacterial and antibiofilm potential against nosocomial bacteria and could serve as an adjuvant in the control of these pathogens.

## 1. Introduction

The incidence of health care-acquired infections, or nosocomial infections (NIs), is a challenging problem in medical practice. The high mortality rates and financial costs of these infections represent a serious problem for health services around the world [1,2,3,4]. Physicians currently face pathogens with resistance determinants that severely restrict therapeutic options; the genetic plasticity of microbes allows them to adapt to stressors through mutations, the acquisition or exchange of genetic material, and the modulation of gene expression, making them resistant to any antimicrobial used in clinical practice [5,6,7,8]. The evolution of hypervirulent strains [9,10,11,12,13], as well as the transmission of and increase in microorganisms with resistance genes, including New Delhi strains [1,14,15] and ESKAPE pathogens (*Enterococcus faecium*, *Staphylococcus aureus*, *Klebsiella pneumoniae*, *Acinetobacter baumannii*, *Pseudomonas aeruginosa* and *Enterobacter* spp.), increase the burden of disease and mortality rates due to treatment failure, representing a global threat to human health [16,17]. According to the World Health Organization (WHO), the global burden of NIs ranges between 7% and 12% [4]. In the United States, these infections are an important factor in patient morbidity, constituting the sixth leading cause of death, surpassing deaths from AIDS, cancer and traffic accidents [18]; Each year, it is estimated that more than 2 million infections are caused by antimicrobial-resistant pathogens, with 29,000 deaths [16]. The European Center for Disease Control and Prevention reported that more than 81,000 patients suffer from NIs daily, while these infections cause approximately 150,000 deaths per year [19]. In 2017, a study of the global burden of disease estimated there to be 48.9 million cases of sepsis worldwide, causing 11 million deaths that same year. The highest burden of sepsis was seen in low- and middle-income countries and accounted for 85% of all sepsis-related deaths worldwide; sepsis cases in India alone were estimated at 11.3 million, with 2.9 million deaths, which is equivalent to 297.7 per 100,000 population [20]. ESKAPE pathogens are responsible for the main cases of NIs worldwide; therefore, the WHO has placed them on the list of bacteria against which the development of new antibiotics is vital [21].

Gram-negative bacteria represent the main driver of NIs; some of these bacteria are naturally resistant to certain families of antibiotics, while others, when subjected to prolonged use, eventually develop resistance, worsening the prognosis of patients [22]. People who are immunocompromised or who are undergoing invasive medical treatment are more vulnerable to infections by this type of pathogen. This growing threat to human health stimulates our interest in the search for new compounds with antimicrobial potential, which are effective and safe for the host and can lead to innovative strategies and allow the development of new options and/or antimicrobial therapies for their control.

In this scenario, natural products and their structural analogs have historically made significant contribution to pharmacotherapy, especially in infectious diseases and cancer [23], moving around USD 20 billion in the global pharmaceutical market each year [24,25]. As a source of specialized metabolites with recognized medicinal properties, compounds of plant origin represent an excellent alternative [26]. They can be directly used as bioactive compounds, drug prototypes and/or as pharmacological tools for different targets [27]. The clinical relevance of monoterpenes has been extensively studied; its wide spectrum of biological and therapeutic activity [28,29,30,31], especially the antimicrobial potential of thymol, linalool, citral and carvacrol has been demonstrated [32,33,34,35,36,37,38,39,40,41,42]. Various studies have related the antimicrobial activity of monoterpenes with their chemical structure, indicating that their lipophilicity facilitates the penetration of pathogens into the cell membrane [38], and their broad spectrum of action has been attributed to the hydroxyl substituent present in their structure [43,44]. 

ISO (2-isopropyl-3,6-dimethoxy-5-methylphenol), a monoterpene first obtained from the aerial parts of *Eupatorium saltense* (Asteraceae) [45], whose synthesis has also been carried out [46], has also been extracted from leaves of *Oxandra xylopioides* (Annonaceae). This compound has been shown to have important biological activities that include: antioxidant [47], anti-inflammatory [48], and antispasmodic [49] effects; vasodilator properties [50]; cryoprotectant effects in canine semen [51]; insecticide activities [52]; and antifungal effects against phytopathogens [53] and human pathogenic yeasts [54,55]. However, despite its significant biological activity, antibacterial potential against human pathogens has not been reported. Therefore, we hypothesize that ISO could have activity against human pathogenic bacteria that cause NIs. This study aimed to evaluate the antibacterial activity of ISO extracted from the leaves of *O. xylopioides*, via a screening of 90 bacterial clinical isolates that included 14 different species, as well as estimating their ability to eradicate mature biofilms of *P. aeruginosa*. The results of this study contribute to the knowledge of the biological potential of this natural compound and further research of novel plant compounds that can be used as adjuvants in the control and treatment of these pathogens.

## 2. Results

### 2.1. Obtaining and Identification of Isoespintanol

ISO was obtained as a crystalline amorphous solid from the petroleum benzine extract of *O. xylopioides* leaves, and its structural identification was performed by GC-MS, ^1^H-NMR, ^13^C-NMR, DEPT, COSY ^1^H-^1^H, HMQC and HMBC. Information related to obtaining and identifying the ISO was reported in our previous study (Supplementary Materials) [54]. 

### 2.2. Antibacterial Susceptibility Testing

ISO showed antibacterial activity against the clinical isolates tested in this study. We observed an inhibition of the growth of bacteria treated with the ISO in comparison with the untreated isolates used as control. Table 1 shows the percentages of inhibition of bacterial growth in the presence of the different concentrations of ISO and commercial antibiotics (ATBs) used as positive controls. The inhibitory effect observed in the clinical isolates was shown to be dependent on ISO concentration; at a higher ISO concentration, we observed a higher percentage of growth inhibition in all clinical isolates. We emphasize that the effect of ISO was different for all isolates, even those belonging of the same species.

Table 2 shows the MIC_90_ and MIC_50_ values of ISO against the evaluated clinical isolates. The greatest effect was observed in isolate A065 of *S. epidermidis*, with MIC_90_ and MIC_50_ values of 694.3 and 154.2 µg/mL, respectively. The smallest effect was observed in *P. aeruginosa* isolate A012, with MIC_90_ and MIC_50_ values of 916.5 and 457.3 µg/mL, respectively.

Figure 1 shows the trend of the data and regression line with 95% confidence. A reduction in the percentage of growth inhibition of bacterial isolates exposed to ISO (MIC of each isolate) is observed, highlighting a strong positive correlation between the concentration of ISO and the percentage of growth inhibition, with Pearson correlation coefficients between 0.86 and 0.96 in most isolates. The hypothesis test on the correlation coefficient yields a *p*-value of <0.05, which indicates a significant linear relationship with 95% confidence. For the isolates of *P. aeruginosa* and *E. coli*, the Spearman test [Rho] (0.96 and 0.97, respectively) was used, which also shows a strong positive relationship between the ISO concentration and the growth reduction of these isolates.

Figure 2 shows the directly proportional relationship between the ISO concentration (µg/mL) and bacterial growth inhibition of each group of clinical isolates.

### 2.3. Biofilm Reduction

All *P. aeruginosa* isolates used in this study were moderate biofilm producers with OD_590_ between 1.52 and 2.79, unlike isolate A050, which was a strong biofilm producer with OD_590_ > 3.0, as shown in Figure 3A. Figure 3B shows the production of *P. aeruginosa* biofilms in the presence of ISO, CIP, and without treatment (INO), evidencing a lower biomass of biofilms when exposed to ISO and CIP compared to cells without treatment. Figure 3C shows the percentage of biofilm eradication of ISO and CIP, highlighting a significantly greater effect of ISO compared to CIP. In the biofilms treated with ISO, a biomass eradication of between 6.6 and 77.4% was obtained after 1 h of exposure. The biomass eradication of biofilms in cells treated with CIP was lower (between 4.3 and 67.5%). It should be noted that isolate A050, which presented a strong biofilm biomass production, was one of the isolates with the highest biofilm eradication (68.2%) by ISO.

## 3. Discussion

The incidence of NIs represents a serious health problem, increasing rates of morbidity, mortality, and costs for health services around the world. An important factor in the increased mortality of NIs is the increasing prevalence of multiresistant microorganisms that render antibiotics ineffective in the treatment of many common infectious diseases [18]. In this context, Gram-positive [10,12], multiresistant Gram negative [11,12] and hypervirulent [9] bacteria are of great concern. For this reason, the search for alternative and novel compounds that have action against these pathogenic microorganisms is becoming increasingly urgent.

In this study, we evaluated the antibacterial activity of ISO against 90 nosocomial isolates, distributed in 14 species that include multiresistant clinical isolates. In this investigation, we report MIC_90_ values between 694.3 and 916.5 µg/mL and MIC_50_ values between 154.2 and 457.3 µg/mL. These results are in agreement with other studies that report the antimicrobial activity of terpenes against a wide variety of microorganisms [37,42,56,57]. However, our results are the first to reveal the antibacterial potential of the natural monoterpene ISO against human pathogenic bacteria. The hydrophobic character of the structure in the cell membranes of microorganisms makes them important targets for the action of monoterpenes; the correlation between the chemical structure of these metabolites and their antimicrobial activity has been described [38,44]. The antibacterial activity of monoterpenes with a chemical structure similar to ISO has been investigated. The treatment of Gram-negative and Gram-positive bacteria with phenolic terpenoids, such as carvacrol and thymol, indicates damage to the integrity of the cell membrane and leakage of intracellular material, highlighting the importance of hydrophobicity and the presence of a phenolic hydroxyl group, disrupting membrane integrity and establishment of its antibacterial activity [28,58]. Similarly, the antifungal activity of these terpenes against pathogenic yeasts has been reported, indicating that their lipophilicity allows interaction with the fungal cell wall, facilitating their penetration into the cell membrane [38]. The monoterpene linalool has also been reported to have antimicrobial activity due to its action in membrane potential, suggesting membrane depolarization, the irregular activity of cell metabolism and damage to the respiratory chain, ultimately leading to cell death [37]. The antifungal action of ISO against pathogenic yeasts of the genus *Candida* was recently described, reporting damage to the cell membrane and the induction of intracellular reactive oxygen species, causing the death of the yeast [54,55]. On the other hand, it has also been reported that bacteria in the presence of compounds with structures similar to ISO, such as thymol, limonene, carvacrol, cinnamaldehyde and eugenol, can modulate the ratio of membrane fatty acids, from saturated to unsaturated; an increase in unsaturated membrane fatty acids and increased fluidity has been reported in the presence of these metabolites, which may affect transport or enzymatic processes at the membrane level; this could be related to the antibacterial action mechanisms of these compounds [59]. All of this indicates that the antibacterial action of ISO could also be associated with damage to the integrity of the bacterial cell membrane.

*Quorum sensing* (QS), is a cell density-based signaling system that aids bacteria-bacteria communication and regulates several virulence factors, including biofilm formation [60]. It is well known that the formation of biofilms is related to the resistance to antimicrobials expressed by pathogenic microorganisms, since they hinder or prevent the penetration of antimicrobials to the site of infection. *Pseudomonas aeruginosa* is known for its ability to form powerful biofilms, which increases its ability to cause a host infection and facilitates the establishment of chronic infections [61,62,63]. Respiratory infection by *P. aeruginosa* is the main cause of morbidity and mortality in patients with cystic fibrosis; biofilm formation in the respiratory tract is thought to increase persistence and resistance to antibiotics during infection [64]. Taking this into account, we also evaluated the ability of ISO to eradicate mature biofilms in this pathogen. All *P. aeruginosa* isolates in this study were biofilm producers. We highlight the role of ISO in the eradication of mature bacterial biofilms during 1 h of treatment, showing eradication percentages of between 6.6 and 77.4%. These results are consistent with previous studies that report the action of ISO against mature biofilms of pathogenic yeasts [54,55], most likely by the inhibition of important components of the biofilms formed by these bacteria, as described in [28]. Studies carried out with the essential oil (EO) of *Thymbra capitata*, a compound rich in thymol, showed an inhibition in the swarm motility, aggregation capacity and hydrophobic capacity of *P. aeruginosa*, further indicating a reduction in the production of three virulence factors regulated by the QS system, including pyocyanin, rhamnolipids, and LasA protease [65]. Monoterpenes, such as citral and carvacrol, are also reported to have antibiofilm activity against pathogenic bacteria [66]. On the other hand, the analysis of the structure-activity relationship, carried out with hordenine and its analogs against strains of *P. aeruginosa* and *S. marcescens*, indicates that the hydroxyl group in the benzene ring present in the structure of these compounds is related to its inhibitory activity of QS and the consequent formation of biofilms [67]. It should be noted that ISO also has this hydroxyl group on the benzene ring, which could be related to its ability to eradicate mature biofilms in these pathogens. Comparing the efficacy of ISO and CIP in the eradication of these biofilms, we found that ISO had a greater effect (between 6.6 and 77.4%), being greater than the effect of CIP in all cases (between 4.3 and 67.5%).

In addition to damage to cell membrane integrity, other mechanisms of the antibacterial action of monoterpenoids have been proposed, including the inhibition of efflux pumps, prevention of the formation and rupture of preformed biofilms, inhibition of bacterial motility, and inhibition of membrane ATPases. Furthermore, it was discovered that they can act synergistically with conventional antibiotics to overcome the problem of bacterial resistance [68]. For all the above, it is interesting to continue investigating the mechanisms of antibacterial action expressed by the natural monoterpene ISO.

Our results provide new and important knowledge on the antibacterial and antibiofilm potential of monoterpene ISO against bacteria causing NIs. In addition, these results serve as a basis for future research on the exploration of mechanisms of action of ISO against pathogenic bacteria. 

## 4. Materials and Methods

### 4.1. Reagents

Mueller-Hinton broth (MHB) (Sigma, Mendota Heights, MN, USA) was used for the determination of MIC and cultures of bacterial isolates. Tryptic Soy Agar (TSA) and Tryptic Soy Broth (TSB) (Becton, Dickinson and Company, San Diego, CA, USA), Mueller-Hinton agar (MHA) (Sigma, Mendota Heights, MN, USA), and Brain Heart Infusion (BHI) broth (Sigma-Aldrich, St. Louis, MO, USA) were also used for bacterial cultures. Dimethyl sulfoxide (DMSO), phosphate-buffered saline (PBS), crystal violet (CV) and antibiotics (ATBs): ciprofloxacin (CIP), amikacin (AMK), ampicillin/sulbactam (SAM), gentamicin (GEN), meropenem (MEM), vancomycin (VAN), and trimethoprim/sulfamethoxazole (SXT) used in this study were obtained from Sigma-Aldrich, St. Louis, MO, USA; glacial acetic acid was obtained from Carlo Erba Reagents, Milano, Italy.

### 4.2. Microorganisms

In this study, 90 clinical isolates were evaluated, distributed in 14 species that included: *Escherichia coli* (38), *Pseudomonas aeruginosa* (12), *Klebsiella pneumoniae* (13), *Acinetobacter baumannii* (3), *Proteus mirabilis* (7), *Staphylococcus epidermidis* (3), *Staphylococcus aureus* (5), *Enterococcus faecium* (1), *Enterococcus faecalis* (1), *Stenotrophomonas maltophilia* (2), *Citrobacter koseri* (2), *Serratia marcescens* (1), *Aeromonas hydrophila* (1) and *Providencia rettgeri* (1). Isolates were cultured from tracheal aspirate samples, blood cultures, bronchoalveolar lavage, tissue secretions, surgical wound secretions, bronchial secretions, sputum, abscesses, and urine cultures from patients hospitalized at the Salud Social S.A.S. from the city of Sincelejo, Sucre, Colombia. All microorganisms were identified by standard systems: Vitek^®^ 2 Compact. Biomerieux SA. (AST-P577, AST-N272, AST-GN93, AST-N271, AST-P612). To maintain the bacterial cultures, BHI broth, TSB, TSA, MHB, MHA and blood agar were used.

### 4.3. Antibacterial Susceptibility Testing

The minimal inhibitory concentration (MIC) of the ISO against clinical isolates was defined as the lowest concentration at which 90% (MIC_90_) of bacterial growth was inhibited, compared to the control (untreated cells). The MIC_50_ was defined as the lowest concentration at which 50% of bacterial growth was inhibited. MIC was determined by performing broth microdilution assays, using 96-well microtiter plates (Nunclon Delta, Thermo Fisher Scientific, Waltham, MA, USA), as described in the *Clinical Laboratory Standards Institute* (CLSI) method M07-A9 [69], with minor modifications. Serial dilutions in MHB were made to accurately obtain final concentrations of 19.5, 39.1, 78.1, 156.2, 312.5, 625, and 1000 µg/mL of ISO in each reaction. A stock solution of ISO at 20,000 µg/mL in DMSO was prepared for carrying out the experiments; in addition, stock solutions of the ATBs used as controls were also prepared. The assays were developed at a final volume of 200 µL per well as follows: 100 µL of the bacterial inoculum at a concentration of 10^8^ CFU/mL and 100 µL of the adjusted ISO system to reach the previously described concentrations in a final reaction. Wells with bacterial inoculum, either without ISO or with ATBs (CIP 6 µg/mL, AMK 20 µg/mL, SAM 2 µg/mL, GEN 8 µg/mL, MEM 1 µg/mL, VAN 2 µg/mL, SXT 20 µg/mL) were used as growth controls and positive controls, respectively. Wells with culture media without inoculum and without ISO were used as negative controls. For each experiment, the controls were made with different concentrations of ISO in culture medium without inoculum. The plates were incubated at 37 °C for 24 h. The experiments were performed in triplicate. The inhibition of bacterial growth by ISO was determined by changes in optical density using a SYNERGY LX microplate reader (Biotek), at 600 nm, from the start of incubation to the end of incubation (24 h). Finally, the percentage of inhibition of bacterial growth was calculated [70] using the following equation: %Inhibition = (1 − (OD_t24_ − OD_t0_/OD_gc24_ − OD_gc0_)) × 100
where OD_t24_: optical density of the test well at 24 h post-inoculation; OD_t0_: optical density of the test well at 0 h post-inoculation; OD_gc24_: optical density of the growth control well at 24 h post-inoculation; OD_gc0_: optical density of the growth control well at 0 h post-inoculation.

### 4.4. Quantitative Assessment of Biofilm Formation

Clinical isolates of *P. aeruginosa* were used as a model to quantify biofilm reduction caused by ISO following the reported methodology [71], with minor modifications. For the formation of biofilms, bacterial colonies of 24 h of incubation in TSA were used, standardizing the bacterial inoculum at 10^8^ cell/mL. Then, in 96-well polystyrene microplates, 200 µL of the bacterial inoculum was discharged into each well and incubated at 37 °C for 24 h. Subsequently, the broth was removed from the microplates, and 200 µL of ISO was added to the MIC of each isolation in TSB broth and incubated at 37 °C for 1 h. Then, the floating cells were removed, and the biofilms at the bottom of the wells were washed three times with deionized water. Excess moisture was removed by tapping the microplates on sterile napkins, and the plates were dried for 5 min. Three assays were performed, and each isolate was tested in 6 replicates. Cultures without ISO were used as control, and CIP was used as positive control. Biofilm reductions were quantified by staining wells with 200 µL of 0.1% CV for 20 min. The samples were washed with deionized water until the excess dye was removed; the excess of water was carefully dried, and then the CV was solubilized in 250 µL of 30% glacial acetic acid. Absorbance values were measured at 590 nm (OD_590_), using a SYNERGY LX microplate reader (Biotek). Biofilm production was grouped into the following categories: OD_590_ < 0.1: non-producers (NP), OD_590_ 0.1–1.0: weak producers (WP), OD_590_ 1.1–3.0: moderate producers (MP) and OD_590_ > 3.0: strong producers (SP). Biofilm reduction was calculated [72] using the following equation:% Biofilm reduction: AbsCO − AbsISO/AbsCO × 100
where AbsCO: absorbance of the control sample and AbsISO: absorbance of the sample treated with ISO.

### 4.5. Statistical Analysis

The data were analyzed using the statistical software R version 4.1.1. (*R Development Core Team*, 2021, Copenhagen, Denmark) and the Excel program. In principle, the Shapiro–Wilk test was used to determine the distribution of data. Subsequently, the Pearson correlation coefficient (for most of the isolates) and Spearman’s test (for *P. aeruginosa* and *E. coli*) were used to measure the degree of linear correlation between the ISO concentration and the reduction in bacterial growth. To compare the effects of ISO and CIP on the reduction in the biofilms, Tukey’s test was used. All experiments were performed in triplicate.

## 5. Conclusions

In this study, we investigated the antibacterial effect of ISO against 90 clinical isolates, as well as its role in biofilm eradication in *P. aeruginosa*. Our results show an inhibition of the growth of the bacteria treated with ISO, in comparison with the untreated isolates used as controls. The inhibitory effect was dependent on ISO concentration and different for all isolates. We also highlight a significantly greater effect of ISO compared to CIP in eradicating mature *P. aeruginosa* biofilms. The antibacterial potential of ISO against these pathogens is demonstrated in this study.

## Figures and Tables

**Figure 1 molecules-27-08004-f001:**
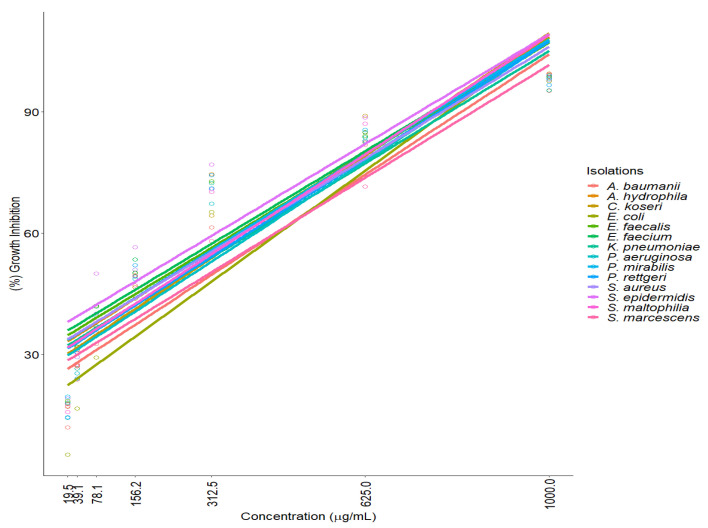
Strong positive correlation between the concentration of ISO and the percentage of growth inhibition of clinical isolates. We observed that the higher the ISO concentration, the higher the inhibition of microbial growth. The hypothesis test on the correlation coefficient with a *p*-value < 0.05, indicates that there is a significant linear relationship, with 95% confidence.

**Figure 2 molecules-27-08004-f002:**
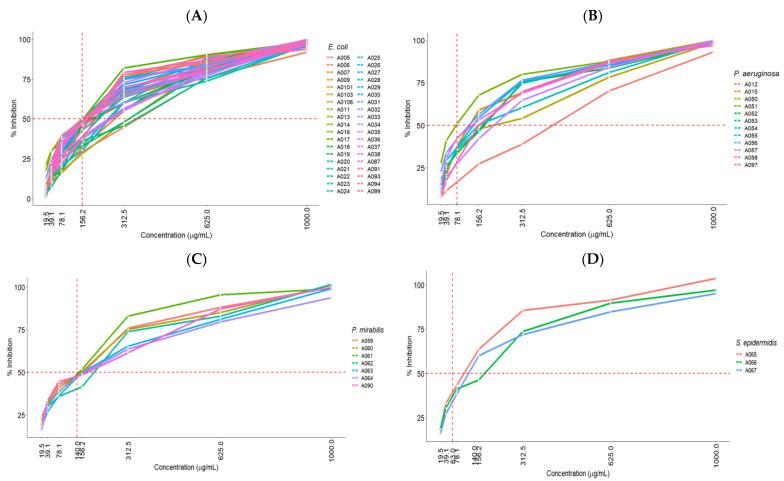
Percentages of growth inhibition showing the MIC_50_ of each group of species of clinical isolates at different ISO concentrations—(**A**): *E. coli*; (**B**): *P. aeruginosa*; (**C**): *P. mirabilis*; (**D**): *S. epidermidis*; (**E**): *A. baumannii*; (**F**): *K. pneumoniae*; (**G**): *S. aureus*; (**H**): *Enterococcus* (*E. faecium*, *E. faecalis*); (**I**): (*C. koseri*, *S. marcescens*, *A. hydrophila*, *S. maltophilia*, *P. rettgeri*).

**Figure 3 molecules-27-08004-f003:**
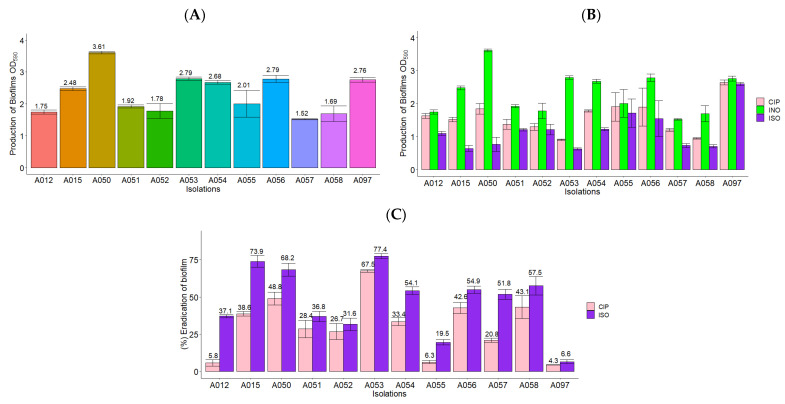
ISO and CIP action on *P. aeruginosa* biofilms. (**A**) Biofilm formation at 37 °C for 24 h.The OD_590_ between 1.1. and 3.0 indicates moderate biofilm production and OD_590_ > 3 indicates strong biomass production in biofilms. (**B**) Biofilm production in the presence of ISO, CIP and without treatment (INO). (**C**) Percentage of biofilm eradication after 1 h of treatment with the MIC of ISO and CIP for each isolate. The results of the ANOVA have a value of *p* < 0.05 and Tukey’s test has a confidence level of 95%, indicating that there is a significant difference between the effect of ISO and the effect of CIP on the eradication of biofilms.

**Table 1 molecules-27-08004-t001:** Percentages of growth inhibition of clinical isolates at different ISO concentrations.

Isolates	ISO Concentrations (µg/mL)							
	19.5	39.1	78.1	156.2	312.5	625	1000	ATBs
*E. coli*								
A017 *	4.5 ± 1.3	14.6 ± 0.8	24.9 ± 3.0	49.5 ± 9.8	82.0 ± 8.6	90.2 ± 5.9	98.1 ± 0.9	96.0 ± 3.5
A018 *	1.4 ± 0.9	13.8 ± 1.6	17.0 ± 7.4	47.3 ± 5.1	60.1 ± 1.3	80.2 ± 5.5	98.3 ± 1.8	92.0 ± 2.6
A019 *	1.4 ± 2.0	13.8 ± 4.0	17.0 ± 4.0	47.3 ± 6.2	60.1 ± 8.4	80.2 ± 3.1	98.3 ± 0.9	92.0 ± 2.3
A020 *	6.4 ± 1.3	14.5 ± 1.2	25.5 ± 3.2	32.8 ± 4.5	48.0 ± 3.3	80.8 ± 4.2	97.5 ± 1.9	98.6 ± 3.0
A021 *	2.9 ± 1.6	7.2 ± 2.4	17.5 ± 2.5	36.3 ± 4.1	45.6 ± 5.2	75.8 ± 7.7	96.7 ± 2.3	98.7 ± 1.9
A022	6.4 ± 1.7	14.4 ± 7.1	21.6 ± 3.2	39.2 ± 2.0	66.3 ± 9.5	79.1 ± 2.3	95.5 ± 3.6	97.7 ± 1.9
A023	5.3 ± 2.3	19.7 ± 1.8	28.7 ± 3.9	33.4 ± 8.0	60.5 ± 3.5	73.7 ± 2.1	96.3 ± 3.3	95.9 ± 1.1
A031	6.4 ± 2.4	15.8 ± 4.3	37.5 ± 3.1	48.7 ± 4.1	74.8 ± 1.7	83.6 ± 3.7	95.2 ± 2.0	95.5 ± 0.5
A035	6.3 ± 2.8	16.6 ± 5.2	32.1 ± 4.5	47.7 ± 2.2	72.7 ± 3.9	87.3 ± 3.8	99.6 ± 0.3	98.9 ± 1.2
A036	2.7 ± 1.8	17.5 ± 4.8	27.7 ± 7.0	38.7 ± 6.0	56.7 ± 8.2	77.2 ± 6.5	99.7 ± 0.3	99.5 ± 0.6
A037 *	2.8 ± 0.2	18.5 ± 3.6	39.8 ± 3.1	48.9 ± 2.9	64.3 ± 1.2	79.7 ± 3.6	99.1 ± 4.5	96.4 ± 2.0
A038	2.5 ± 0.8	16.7 ± 4.1	35.2 ± 4.1	48.9 ± 3.8	67.0 ± 5.1	80.1 ± 1.8	96.7 ± 3.3	97.0 ± 1.1
A007	1.1 ± 1.3	10.7 ± 3.6	19.6 ± 6.6	32.3 ± 8.9	55.0 ± 5.1	84.6 ± 3.1	98.1 ± 0.5	97.8 ± 1.7
A006 *	2.7 ± 1.7	10.4 ± 2.1	15.9 ± 5.8	29.5 ± 3.7	44.8 ± 1.6	76.0 ± 4.3	91.9 ± 6.0	96.7 ± 2.0
A009	6.9 ± 1.2	16.3 ± 6.9	32.3 ± 8.7	46.2 ± 1.3	68.9 ± 4.8	82.7 ± 3.2	97.8 ± 1.5	94.8 ± 2.3
A016	3.5 ± 1.2	15.9 ± 2.4	35.2 ± 8.6	49.8 ± 4.8	63.5 ± 3.0	79.1 ± 4.6	99.1 ± 0.9	98.1 ± 1.0
A087	5.7 ± 3.4	15.1 ± 2.3	33.5 ± 4.3	49.9 ± 4.4	69.5 ± 3.4	82.9 ± 7.0	99.2 ± 0.5	98.8 ± 0.7
A091 *	7.0 ± 2.2	16.6 ± 6.5	30.6 ± 6.5	49.6 ± 2.8	69.9 ± 6.7	80.9 ± 2.9	97.5 ± 2.1	98.7 ± 1.6
A093	5.4 ± 2.7	16.1 ± 1.5	25.4 ± 1.7	43.2 ± 5.1	79.1 ± 5.7	86.9 ± 1.5	93.8 ± 3.3	94.1 ± 3.8
A094 *	3.8 ± 1.3	14.2 ± 0.7	24.5 ± 1.9	37.3 ± 6.2	77.6 ± 4.3	88.9 ± 8.3	96.2 ± 2.6	98.2 ± 2.1
A099	1.2 ± 0.5	22.2 ± 6.0	39.9 ± 4.8	46.0 ± 1.4	77.6 ± 1.0	85.2 ± 2.9	98.8 ± 1.8	98.3 ± 1.3
A0101	2.2 ± 0.1	15.9 ± 3.5	28.1 ± 1.7	49.9 ± 5.8	75.0 ± 3.8	89.8 ± 2.1	98.0 ± 2.2	98.8 ± 1.3
A0103 *	1.0 ± 0.5	16.1 ± 0.5	39.2 ± 1.9	47.4 ± 2.6	68.5 ± 1.8	85.9 ± 5.7	97.3 ± 2.5	98.9 ± 2.6
A0106 *	3.3 ± 1.4	14.8 ± 2.5	37.3 ± 3.4	49.0 ± 2.8	76.0 ± 2.1	86.7 ± 2.3	98.9 ± 2.5	98.9 ± 1.3
A024	3.6 ± 1.5	19.3 ± 4.5	28.9 ± 2.5	42.9 ± 3.9	63.7 ± 5.3	81.4 ± 3.7	97.3 ± 2.4	99.9 ± 1.7
A025 *	1.6 ± 1.1	13.4 ± 7.7	20.5 ± 4.2	30.3 ± 4.4	60.2 ± 4.5	88.2 ± 1.8	98.1 ± 1.1	98.1 ± 1.1
A026	5.3 ± 1.7	16.3 ± 2.5	29.6 ± 4.0	49.6 ± 5.0	68.9 ± 4.5	75.3 ± 1.1	97.1 ± 1.8	97.2 ± 1.5
A027 *	4.8 ± 1.0	19.2 ± 0.8	34.4 ± 1.9	48.8 ± 3.1	76.0 ± 3.9	83.5 ± 2.5	99.2 ± 0.7	99.8 ± 0.5
A028	4.7 ± 0.9	16.7 ± 1.7	33.7 ± 7.4	47.1 ± 3.2	68.0 ± 2.1	83.2 ± 3.8	97.7 ± 1.2	99.4 ± 0.4
A029 *	5.8 ± 1.4	11.6 ± 3.3	33.7 ± 4.5	47.7 ± 3.8	71.5 ± 2.8	84.1 ± 4.4	99.3 ± 0.2	99.2 ± 0.5
A030	0.6 ± 0.5	17.7 ± 1.5	28.5 ± 0.9	47.6 ± 1.9	69.8 ± 2.5	81.1 ± 4.0	98.7 ± 1.6	98.7 ± 1.0
A032 *	12.0 ± 2.4	24.1 ± 3.2	37.7 ± 2.6	49.5 ± 4.4	64.7 ± 4.5	84.6 ± 4.5	99.0 ± 1.2	99.1 ± 2.5
A033 *	4.6 ± 1.1	19.9 ± 5.5	30.9 ± 1.5	49.9 ± 5.5	60.0 ± 3.7	78.2 ± 2.2	96.7 ± 2.6	99.6 ± 3.3
A034	7.3 ± 1.1	15.4 ± 2.1	21.4 ± 5.7	37.3 ± 1.1	55.4 ± 3.4	80.1 ± 3.1	98.5 ± 1.7	97.8 ± 2.1
A005	6.1 ± 0.9	16.4 ± 0.8	25.5 ± 2.0	46.1 ± 0.9	59.7 ± 3.9	81.1 ± 1.7	95.7 ± 1.3	98.1 ± 2.1
A011	8.4 ± 1.5	16.2 ± 1.4	16.9 ± 2.0	28.3 ± 0.5	47.9 ± 0.9	82.9 ± 4.0	97.6 ± 1.1	98.5 ± 2.1
A013	17.2 ± 0.5	29.5 ± 2.2	38.9 ± 3.6	50.0 ± 6.9	68.7 ± 5.0	82.8 ± 4.7	97.9 ± 2.5	98.9 ± 1.3
A014	20.7 ± 4.3	30.1 ± 2.0	35.0 ± 6.8	49.7 ± 1.2	65.1 ± 2.7	79.6 ± 2.5	98.1 ± 3.8	98.7 ± 1.1
*P. aeruginosa*								
A012	7.9 ± 2.5	11.5 ± 0.8	16.6 ± 4.7	27.2 ± 2.2	38.9 ± 5.2	70.0 ± 8.6	93.0 ± 6.3	99.4 ± 0.4
A015	16.9 ± 4.9	21.0 ± 6.7	37.9 ± 10.7	58.8 ± 4.7	69.2 ± 4.9	88.1 ± 4.2	96.3 ± 2.6	98.9 ± 0.7
A050	13.3 ± 4.5	26.4 ± 2.3	33.5 ± 8.6	47.8 ± 1.9	54.1 ± 1.6	78.1 ± 6.0	98.3 ± 2.0	97.7 ± 2.7
A051	27.4 ± 5.2	40.8 ± 5.4	50.8 ± 3.7	67.7 ± 4.9	79.8 ± 1.4	87.4 ± 1.6	98.1 ± 3.0	96.8 ± 1.1
A052	11.9 ± 5.5	25.3 ± 2.7	33.8 ± 3.3	46.3 ± 5.0	75.3 ± 7.1	83.6 ± 5.6	98.2 ± 2.0	99.3 ± 1.4
A053	8.9 ± 2.9	23.9 ± 5.7	35.7 ± 5.2	47.5 ± 5.7	74.7 ± 1.3	86.2 ± 6.9	99.4 ± 1.0	98.5 ± 3.2
A054	18.4 ± 3.3	24.2 ± 3.3	36.7 ± 0.8	50.3 ± 6.0	60.4 ± 3.2	81.0 ± 2.9	99.2 ± 0.3	100.2 ± 1.4
A055	14.3 ± 4.1	29.4 ± 1.2	37.1 ± 6.0	54.4 ± 1.0	75.7 ± 2.5	85.0 ± 2.6	99.3 ± 2.7	98.6 ± 3.2
A056	22.4 ± 1.4	33.7 ± 6.7	42.0 ± 8.3	56.2 ± 8.2	76.4 ± 3.5	86.9 ± 3.6	99.2 ± 5.1	99.9 ± 0.6
A057	9.3 ± 1.6	19.5 ± 1.0	27.4 ± 3.0	42.0 ± 1.3	64.6 ± 3.6	83.9 ± 4.2	99.6 ± 1.2	94.3 ± 0.7
A058	13.5 ± 2.8	30.7 ± 3.4	42.6 ± 5.9	53.1 ± 2.1	68.8 ± 2.9	85.7 ± 4.6	98.2 ± 1.5	96.2 ± 2.1
A097	7.9 ± 4.8	17.3 ± 3.1	28.8 ± 0.9	50.1 ± 0.7	69.7 ± 1.6	87.5 ± 1.7	96.9 ± 4.0	95.1 ± 0.9
*P. mirabilis*								
A059	23.7 ± 4.4	32.0 ± 1.1	42.8 ± 1.1	49.3 ± 7.7	75.9 ± 4.1	88.0 ± 5.0	99.6 ± 2.1	98.1 ± 2.7
A060	21.1 ± 2.5	30.1 ± 2.6	40.9 ± 5.7	50.4 ± 2.0	75.1 ± 3.6	85.0 ± 2.7	99.7 ± 1.3	99.3 ± 1.5
A061	20.0 ± 1.9	32.0 ± 3.4	38.3 ± 1.7	51.7 ± 1.2	82.8 ± 5.8	95.4 ± 1.6	98.7 ± 3.2	95.5 ± 3.7
A062	19.1 ± 4.2	29.7 ± 2.6	35.7 ± 3.8	41.5 ± 3.3	73.7 ± 2.5	82.9 ± 2.4	99.7 ± 0.3	94.8 ± 2.2
A063	18.3 ± 0.9	26.7 ± 2.1	36.2 ± 1.5	49.8 ± 1.7	65.1 ± 1.8	81.0 ± 4.9	99.0 ± 3.9	100.0 ± 2.9
A064	16.2 ± 2.8	30.1 ± 2.8	38.8 ± 2.6	49.3 ± 5.4	63.5 ± 4.5	79.3 ± 3.7	93.6 ± 10.0	99.4 ± 1.0
A090	18.7 ± 1.7	33.3 ± 3.9	44.6 ± 1.2	48.6 ± 2.6	61.5 ± 5.6	86.9 ± 7.5	99.7 ± 1.7	100.1 ± 1.7
*S. epidermidis*								
A065	18.5 ± 0.7	33.3 ± 2.5	43.1 ± 3.7	63.4 ± 3.3	85.4 ± 2.0	91.4 ± 4.2	99.9 ± 0.3	99.7 ± 1.4
A066	18.7 ± 1.2	30.4 ± 1.9	41.3 ± 1.7	46.2 ± 1.8	73.6 ± 2.2	89.6 ± 3.4	97.0 ± 0.5	98.4 ± 0.4
A067	16.0 ± 2.0	27.2 ± 5.3	38.4 ± 5.8	59.9 ± 1.4	71.8 ± 2.2	84.7 ± 6.7	95.0 ± 2.4	96.8 ± 1.6
*A. baumannii*								
A001	5.0 ± 2.0	14.0 ± 0.6	26.5 ± 1.7	38.6 ± 3.0	55.0 ± 1.9	75.2 ± 0.7	91.4 ± 0.6	98.5 ± 1.0
A070	13.2 ± 2.2	29.5 ± 3.6	35.4 ± 1.2	48.4 ± 1.8	65.5 ± 5.7	89.2 ± 4.2	95.7 ± 1.8	94.8 ± 1.2
A089	17.8 ± 1.5	27.9 ± 2.5	35.9 ± 0.8	46.6 ± 2.5	63.5 ± 2.1	81.1 ± 7.0	98.4 ± 2.5	96.8 ± 2.8
*K. pneumoniae*								
A008	25.7 ± 6.0	38.7 ± 2.1	43.1 ± 1.8	51.7 ± 1.2	67.3 ± 1.7	72.8 ± 3.0	96.9 ± 1.0	99.0 ± 1.6
A039	24.3 ± 2.6	31.4 ± 4.8	43.3 ± 4.5	55.3 ± 5.6	68.8 ± 2.7	79.7 ± 6.3	93.9 ± 3.6	99.9 ± 0.5
A040	17.2 ± 3.1	26.4 ± 1.9	34.1 ± 5.8	42.8 ± 7.1	73.4 ± 3.9	80.0 ± 0.8	90.3 ± 4.8	100.5 ± 1.5
A041	16.9 ± 2.4	24.1 ± 8.4	39.6 ± 5.5	48.8 ± 2.3	78.2 ± 1.6	89.2 ± 4.4	99.7 ± 2.5	97.8 ± 4.2
A042	18.7 ± 0.8	27.3 ± 5.7	33.3 ± 4.6	47.6 ± 6.5	80.5 ± 2.7	87.2 ± 3.4	91.0 ± 1.9	97.4 ± 3.2
A043+	10.8 ± 3.7	21.4 ± 2.8	36.3 ± 4.2	49.6 ± 7.9	79.8 ± 2.8	82.8 ± 4.5	98.9 ± 4.3	95.8 ± 0.7
A044	16.9 ± 1.6	27.6 ± 8.3	41.7 ± 2.7	48.9 ± 1.3	74.1 ± 2.6	85.8 ± 3.0	98.2 ± 3.2	97.6 ± 4.4
A045	16.2 ± 1.2	20.1 ± 2.6	34.0 ± 3.4	55.1 ± 7.7	69.4 ± 6.7	85.3 ± 6.7	99.4 ± 1.2	97.1 ± 0.6
A046	13.7 ± 2.3	16.7 ± 1.5	28.6 ± 5.8	45.3 ± 7.1	72.7 ± 5.8	89.6 ± 3.2	97.6 ± 7.5	99.9 ± 1.1
A047	24.1 ± 3.6	26.6 ± 4.4	38.2 ± 0.7	50.5 ± 0.9	67.8 ± 8.3	74.3 ± 3.9	90.7 ± 3.0	99.3 ± 2.2
A048	19.9 ± 4.8	30.1 ± 1.6	40.0 ± 1.3	47.7 ± 5.5	66.3 ± 10.8	79.7 ± 2.0	92.1 ± 1.8	99.2 ± 0.7
A049	17.5 ± 3.5	28.1 ± 4.8	35.3 ± 3.8	46.7 ± 3.3	74.5 ± 3.9	89.0 ± 5.4	91.6 ± 4.9	99.1 ± 0.8
A0104+	12.4 ± 3.1	26.5 ± 1.4	33.4 ± 6.0	49.8 ± 5.1	68.1 ± 4.4	81.4 ± 2.5	98.6 ± 2.9	100.3 ± 1.7
*S. aureus*								
A004	17.6 ± 4.9	25.8 ± 1.7	40.7 ± 0.8	48.8 ± 2.5	67.3 ± 6.9	77.9 ± 4.5	98.9 ± 2.0	99.3 ± 2.1
A010	21.2 ± 0.2	29.3 ± 3.1	41.4 ± 8.5	58.8 ± 3.5	69.4 ± 3.2	84.2 ± 1.9	98.6 ± 1.2	94.4 ± 7.0
A072	22.7 ± 1.2	32.7 ± 1.0	42.6 ± 3.2	49.2 ± 1.9	71.7 ± 2.4	88.6 ± 6.6	99.1 ± 1.4	99.5 ± 2.2
A073	13.8 ± 1.4	28.0 ± 0.1	35.8 ± 5.1	49.8 ± 2.6	68.1 ± 5.5	75.2 ± 4.5	93.2 ± 3.0	98.8 ± 1.1
A0100	20.5 ± 1.2	30.8 ± 3.1	40.1 ± 2.5	48.4 ± 2.7	77.7 ± 0.7	87.1 ± 2.6	99.6 ± 5.1	98.5 ± 1.3
*E. faecium*								
A0105	20.6 ± 6.3	32.2 ± 4.0	41.7 ± 5.7	56.8 ± 2.3	75.8 ± 1.1	85.9 ± 1.2	98.7 ± 2.5	99.4 ± 1.1
*E. faecalis*								
A069	18.5 ± 1.7	31.8 ± 3.4	42.1 ± 3.5	50.2 ± 0.6	72.8 ± 2.4	84.0 ± 6.5	98.6 ± 2.3	99.2 ± 1.2
*C. koseri*								
A068	17.1 ± 1.3	29.6 ± 0.7	36.9 ± 3.6	47.2 ± 1.2	73.8 ± 3.0	89.8 ± 1.1	97.4 ± 1.8	99.4 ± 1.0
A079	17.2 ± 5.9	25.1 ± 2.0	36.2 ± 3.9	46.1 ± 0.8	54.9 ± 1.1	88.3 ± 4.8	99.4 ± 0.2	99.9 ± 0.7
*S. marcescens*								
A071	17.1 ± 0.6	27.1 ± 2.7	35.8 ± 1.7	47.1 ± 3.1	58.1 ± 2.3	71.5 ± 2.8	99.3 ± 0.5	95.6 ± 2.8
*A. hydrophila*								
A088	17.8 ± 0.8	31.1 ± 0.6	36.6 ± 0.5	49.6 ± 6.0	74.6 ± 1.7	84.9 ± 4.2	99.1 ± 1.4	98.5 ± 2.4
*S. maltophilia*								
A0102	17.1 ± 1.3	25.9 ± 2.5	40.9 ± 3.7	46.5 ± 2.3	67.0 ± 3.6	87.4 ± 4.7	98.7 ± 2.7	99.8 ± 0.5
A095	14.4 ± 1.0	29.2 ± 1.3	35.7 ± 1.5	52.5 ± 2.5	73.4 ± 1.7	86.7 ± 1.6	98.4 ± 3.6	98.4 ± 2.2
*P. rettgeri*								
A096	14.5 ± 1.9	24.1 ± 1.8	36.9 ± 3.4	52.1 ± 3.4	74.3 ± 2.6	84.0 ± 2.6	96.6 ± 1.9	99.3 ± 1.0

* BLEE (extended spectrum beta-lactamases); + multi-resistant; ATBs: (AMK 20 µg/mL: A006, A017, A018, A019, A020, A021, A025, A027, A029, A032, A033, A037, A091, A094, A0103, A0106), (GEN 8 µg/mL: A009, A016, A0104), (VAN 2 µg/mL: A065, A066, A067), (MEM 1 µg/mL: A043), (SAM 2 µg/mL: A089), (SXT 20 µg/mL: A095, A0102), (CIP 6 µg/mL was used for the rest of the isolates).

**Table 2 molecules-27-08004-t002:** MIC_90_ and MIC_50_ values of ISO against bacterial clinical isolates.

Isolates	Isoespintanol µg/mL		Isolates	Isoespintanol µg/mL	
	MIC_90_	MIC_50_		MIC_90_	MIC_50_
*E. coli*			*P. aeruginosa*		457.3
A017 *	731.6	296.8	A012	916.5	
A018 *	798.8	367.4	A015	765.3	247.4
A019 *	783.2	358.3	A050	826.2	315.4
A020 *	828.7	387.9	A051	727.3	83.56
A021 *	849.4	422	A052	767.9	275.2
A022	816.2	362.4	A053	749.7	273.4
A023	846.8	372.9	A054	796.7	280.6
A031	782.5	297.3	A055	749.4	235.1
A035	747.5	298.5	A056	740.7	185.8
A036	811.9	365.6	A057	776.8	324
A037 *	788.9	312.9	A058	766.7	237.7
A038	798.2	322.9	A097	766.3	304.4
A007	798.3	389.2	*P. mirabilis*		
A006 *	884.9	442.8	A059	739.4	199.6
A009	781.2	316	A060	751.3	218.3
A016	791.5	324.4	A061	702.9	192.3
A087	768.6	308.6	A062	769.4	259.2
A091 *	786.6	314.6	A063	791.1	269.5
A093	775.8	318.3	A064	843.3	275.7
A094 *	760	329.6	A090	769.5	235.6
A099	750.6	283.4	*S. epidermidis*		
A0101	742.5	304	A065	694.3	154.2
A0103 *	770.4	309.3	A066	756.1	230.1
A0106 *	745.7	292.3	A067	783.9	233.7
A024	799.4	339.4	*A. baumannii*		
A025 *	781.1	373	A001	878.7	394.6
A026	811.6	331.6	A070	778.9	269
A027 *	756.6	291.7	A089	799.6	278.6
A028	780.3	316	*K. pneumoniae*		
A029 *	761.1	312.1	A008	852.2	212.5
A030	777.6	327.5	A039	831.4	215.1
A032 *	774.7	280.1	A040	849.1	289.6
A033 *	818.4	335.1	A041	730.3	235.9
A034	812.7	372.7	A042	795.2	248.5
A005	815.1	350.2	A043+	755.6	266.3
A011	824.3	397.2	A044	760.1	237.5
A013	784.6	251.3	A045	757.9	266.2
A014	805.9	263.2	A046	754.4	294.9
*S. aureus*			A047	891.1	268.8
A004	800.6	266.6	A048	852.3	263.7
A010	767	213.6	A049	794.5	254.8
A072	748.4	209.2	A0104+	784.6	281.9
A073	853.7	289.8	*E. faecium*		
A0100	740.7	216.8	A0105	748.1	196.5
*C. koseri*			*E. faecalis*		
A068	752.3	241.8	A069	767.1	225.1
A079	778	287.9	*S. maltophilia*		
*S. marcescens*			A0102	764.7	257.8
A071	843.9	306.7	A095	754.5	244.8
*A. hydrophila*			*P. rettgeri*		
A088	757.9	236.8	A096	774.6	258

* BLEE (extended spectrum beta-lactamases); + multi-resistant.

## Data Availability

The data presented in this study are available in the article and the supporting information.

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
