# Peer review of "Antibacterial Screening of Isoespintanol, an Aromatic Monoterpene Isolated from Oxandra xylopioides Diels"

_molecules, 2022, doi:10.3390/molecules27228004_

Round 1
Reviewer 1 Report
it is a good work
I recommended he's publication
Reviewer 2 Report
In line number 36 "recent years of pathogens with resistance" add bacterial pathogens instead on pathogens only as authors have targeted only bacterial pathogenesis.
In line number 39-40, authors need to add the burden of disease and mortality rate data globally as well as in Indian scenario.
Add the concrete studies on monoterpene for clinical relevance. Aa introduction is very generous. Authors must be rewrite it with higher relevance to title.
How authors are saying "all clinical isolates". Authors need to add some backed data or studies of studied isolates for clinical claims.
Authors need to add the isolates name of all the clinical isolates used in the study.
Reviewer 3 Report
Dear Authors, the presented study is focused on an interesting and current topic. The study is focused on monitoring the antibacterial effect of isoespintanol against a wide range of bacterial strains. D8le was evaluated for its effect on the reduction of the biofilm produced by Ps. aeruginosa. A considerable number of experiments were carried out as part of the study, but unfortunately the study is not written in a proper and precise form. There are many inconsistencies in the text, language proofreading by native English speakers will also be necessary. Below I attach some detailed comments (but certainly not all discrepancies are listed, I recommend going through the entire text in detail based on some of the comments written here): 1/ L. 20 - values ​​in % are recommended to 1 decimal place 2/ L. 20 -21 - nunto rewrite, word order, English 3/ L. 21 - (CIP) - the abbreviation is not used in the abstract, I consider the introduction of the abbreviation redundant. 4/ L. 31 "doctor" - inappropriate expression, please formulate it better 5/ L. 36 etc. inappropriate word order of sentences, rewrite 6/ Overall, it is necessary to revise the text of the entire manuscript - many sentences need to be simplified, divided into several sentences. It is not an exception when 1 sentence is on 4 lines and combines a lot of information that should be separated (see e.g. L. 84-87, abstract, etc.) 7/ L. 74 - there is no specification of what kind of biofilm it was - P. aeruginosa biofilm? 8/ Chapter 2.1 - the sentence structure of the text is not suitable, the chapter should ideally contain more than 1 sentence. However, the given text is not even a sentence. 9/ Tab 1 etc - "Isolations" - do you mean "isolates"?? otherwise I don't know how it is meant. 10/ Tables are not clearly processed. 11/ Fig 2 - unreadable objects, graphics need to be improved 12/ Napdis 2.3 - redundant dot 13/ L. 211 - quorum sensing - write in italics 14/ L. 266 redundant parenthesis 15/ L. 275 - "y" - ??? 16/ L. 281-282 and other similar sentences in the entire text - please retype and avoid using an unnecessary colon, write as a more readable sentence. 17/ L. 291 MO7-A9 redundant gap 18/ L. 299 etc. - what concentrations of antibiotics were tested??? Dates not given. 19/ L. 309-312 - it is not a sentence (explanation of abbreviations in the equation) and certainly not a separate paragraph of the text... !! 20/ L. 336-338 - see comment 19/
Round 2
Reviewer 2 Report
No comments
Reviewer 3 Report
Unfortunately, the authors did not provide detailed comments on each of my comments (1-20). Please respond to each point of my review and add, for example, the line number where it is modified. Subsequently, I will be able to look at the repairs and improvements made.
